# Comparison of Two *Cutibacterium acnes* Biofilm Models

**DOI:** 10.3390/microorganisms9102035

**Published:** 2021-09-26

**Authors:** Jennifer Varin-Simon, Fabien Lamret, Marius Colin, Sophie C. Gangloff, Céline Mongaret, Fany Reffuveille

**Affiliations:** 1EA 4691 Biomatériaux et Inflammation en Site Osseux (BIOS), Université de Reims Champagne-Ardenne, SFR Cap Santé (FED 4231), 51100 Reims, France; Jennifer.varin-simon@univ-reims.fr (J.V.-S.); fabien.lamret@univ-reims.fr (F.L.); marius.colin@univ-reims.fr (M.C.); sophie.gangloff@univ-reims.fr (S.C.G.); 2UFR Pharmacie, Université de Reims Champagne Ardenne, 51100 Reims, France; 3Service Pharmacie, Centre Hospitalier Universitaire de Reims, 51100 Reims, France

**Keywords:** biofilm models, host-pathogen interactions, *Cutibacterium acnes*

## Abstract

The study of biofilms in vitro is complex and often limited by technical problems due to simplified models. Here, we compared *C. acnes* biofilm formation, from species involved in bone and prosthesis infection, in a static model with a dynamic model. Using similar parameters, the percentage of live bacteria within the biofilm was higher in dynamic than in static approach. In both models, bacterial internalization in osteoblast-like cells, playing the role of stress factor, affected this proportion but in opposite ways: increase of live bacteria proportion in the static model (×2.04 ± 0.53) and of dead bacteria proportion (×3.5 ± 1.03) in the dynamic model. This work highlights the huge importance in the selection of a relevant biofilm model in accordance with the environmental or clinical context to effectively improve the understanding of biofilms and the development of better antibiofilm strategies.

## 1. Introduction

The need for joint prostheses is continuously increasing due to the aging of the population [1,2]. Unfortunately, even taking into account prophylaxis and aseptic surgical techniques, the risk of bone and joint infections does not decrease (approximately 1 to 2% of procedures) [3]. In the event of infection, the presence of a prosthesis allows bacteria to adhere to it, to develop a biofilm and to trigger an infection of the surrounding tissues. This bacterial ability to form a biofilm appears to be a very worrying threat. Resulting from strong social behavior, this microbial community is surrounded by a protective extracellular matrix [3,4,5,6] which is found to be the most effective defense against all types of antimicrobials and against attacks by the immune system [3]. The consequences of biofilm formation are the persistence of these microbial agents which leads to clinical symptoms linked to the inflammatory response such as fever and pain. These sequelae are often irreversible (loosening of prosthesis, amputation), and fatal in some cases. All clinical consequences represent high costs for society [1,7]. In such a clinical context, it becomes urgent to consider new strategies and develop new molecules to treat acute and chronic infections due to biofilms; this goes through a better description of biofilm mechanisms. The structure and composition of the biofilm depend on many parameters: the bacterial strains, the stage of development, the presence or absence of an antibiotic, and especially the microenvironment [8,9,10,11,12,13,14,15,16]. The biofilm will therefore be different depending on the environmental conditions (presence of oxygen, nutrients, and host cells) but also depending on the surface on which it is formed [17,18,19,20]. Thus the proportion of adhered bacteria, living or dead, and the matrix composition within biofilm are different according to the bacterial species and the infected site [21]. All these parameters should be taken in account in experiments.

In consequence, studying biofilm infections is complex, especially because of the limited in vitro models. Many authors agree on the need to jointly use miscellaneous methods to fully describe a biofilm: its structure, organization, and live and dead populations [22,23,24]. Classical crystal violet assay or microscopic visualization thanks to fluorescence dyes are usually used to define biofilm characterizations [25,26,27]. However, in 2019, 29 researchers with international expertise in biofilms came together to identify gaps in this area. The lack of relevant, standardized models adapted to in situ conditions was raised (Biofilm BASH 2019, [23]). Indeed, several publications have shown the huge influence of culture conditions on biofilm formation [28,29,30], but also the influence of the structure and composition of the surface on which the biofilm develops [31,32].

*Cutibacterium acnes* is an emerging anaerobic pathogen frequently involved in prosthetic joint infections. Potential inflammatory and virulent factors involved in adherence to skin tissue were identified among *C. acnes* components: A specific envelope with peptide chain containing L-acid L-diaminopelic acid and D-alanine, and polysaccharide moiety with an amino sugar thought to be a diamunihexuronic acid [33]; lipase leading to neutrophil chemotaxis [34]; other many various factors (neuraminidase, polyunsaturated fatty acid isomerase) and host-interacting factors, such as CAMP factors, hemolysins and dermatan sulphate-binding adhesins (DsA1 and DsA2) [35]. Virulent factors involved in biofilm formation were not clearly identified but adhesins are known to be involved in adhesion, which is the first step of biofilm formation. *C. acnes* infection is evaluated to account for 10% of BPIs but this is probably underestimated [36]. This infection is often delayed (occurring 3 to 24 months or more after prosthesis setting) [37]. The establishment of biofilms explains the complex pathogenesis of *C. acnes*-related BPI. This peculiar bacterial growth makes such chronic infections difficult to diagnose, to treat and to eradicate. In a previous study, several *C. acnes* strains were co-cultured with osteoblast-like cells in order to highlight the impact of the bacterial internalization into the host cells on the biofilm formation of *C. acnes* [38]. We used a static model in microtiter plates, in which biofilm biomass was stained with crystal violet or labelled with fluorochrome markers to quantify live and dead bacteria proportions. However, we were limited for 3D structure observation due to our technical approach. Indeed, the biofilm formed in the microtiter plate was not observable under confocal microscopy due to incompatibility between the focal distance and plastic thickness.

In this study, we hereby used a dynamic flow cell channels model to apprehend confocal microscopic observation using an adapted three-channel flow-cell chamber. Obvious differences appear between the static model and the present dynamic model: the force due to shear flow leading to a physical pressure on biofilm formation, a permanent circulation of nutrients and a variability of oxygen level due to the technic of the flow experiment. In this communication, the differences resulting from both models will be described and compared to underline the importance of choosing the suitable biofilm model to correlate with the conditions of the infectious site studied.

## 2. Methods

### 2.1. Bacteria and Culture

*Cutibacterium acnes* (formerly *Propionibacterium acnes*) strains were isolated at the laboratory of bacteriology of Reims University hospital (CHU Reims). Strains were identified by mass spectrophotometry in routine by University hospital. Four clinical isolates were non-infection related strains (C2, C5, C8, and C18) and after being anonymized, these strains were labeled as infection-naive strains. Nine other *C. acnes* strains were isolated from bone and prosthesis infections BPIs (BPI 1 to BPI 9). They were defined when at least three of the five samples from the bone and joint tissue during orthopedic surgery were positive. *C. acnes* strains were isolated on Columbia agar with 5% sheep blood (BioRad, Hercules, CA, USA) and cultivated in a Brain Heart Infusion (BHI) broth (BioRad, Hercules, CA, USA) for five days under anaerobic conditions using the GenBox system (Biomerieux, Marcy l’Etoile, France) at 37 °C.

### 2.2. Flow Cells Models and Microscopy

In this dynamic model, *C. acnes* biofilms were grown for 96 h at 37 °C in flow chambers (IBI Scientific, Dubuque, Iowa, USA). The system was assembled and sterilized by pumping a 0.2% hypochlorite solution through the system for 2 h using a peristaltic pump. The system was rinsed with sterile water and then by diluted medium (BHI diluted 1/10e with sterile water) for 1 h with a rate of 20 mL/h. Flow chambers were inoculated by injecting a 5-day-old bacterial culture diluted at 1/20. Inoculated chambers were flipped upside-down, without flow for 3 h to let bacteria adhere. Then, fresh diluted medium was pumped through the system at a constant rate of 0.2 mL/h during four days. The flow cells model was detailed in Figure 1. After 96 h, biofilms were stained using the Live/Dead BacLight bacterial viability kit (Molecular probes) at a ratio 1:5 of Syto-9 to propidium iodide. Biofilms were observed using a LMS 710 NLO confocal laser scanning microscope (Zeiss), and two joined fields were captured. The observation of the biofilm was carried out twice for the strains C2, C5, C8, and C18 after internalization and once for the other experiments. The homogeneity of the samples was checked by traversing the observation field and the most representative area was chosen for the acquisition of the image. Three-dimensional reconstructions and fluorescents volumes were generated using Imaris software.

## 3. Results and Discussion

Based on our previous results [38], after testing seven commensal *C. acnes* strains, four were selected for their capacity to increase biofilm formation after their internalization in osteoblast-like cells to perform our experimentations. This change in their biofilm capacity served as a “landmark” element for comparison between static and dynamic models. Thus, we evaluated a dynamic flow model (Figure 1), which couples the use of flow cells and confocal microscopy to appreciate the structure and organization of biofilm (Figure 2a).

The four strains presented biofilms of equal volume (Figure 2b), however, differences should be noted. The C2 and C5 strains formed a thick monolayer biofilm, which was highly homogeneous, however, with aggregates more present in C5 isolate. Strain C8 also formed a monolayer biofilm but the aggregates were more dispersed, and biofilm appeared thinner. C18 strain formed a biofilm, which was completely different with large and very thick aggregates. Between these “mushroom”-type formations often described in other species, we observed a monolayer biofilm but shaped by as many dead bacteria as living. Thus, this revealed organizational differences between strains, with thick monolayer biofilm (C2, C5, C8) or the presence of larger aggregates and different proportion of dead bacteria (C18).

For different purposes, we evaluated the proportion of living and dead bacteria (Figure 2c) within biofilm. First, if the biofilm is mainly composed of living bacteria, these are cells that might be released much later and therefore induce new contaminations on other sites [3]. The second reason is that the quantity of dead bacteria can also release DNA after lysis, which participates in matrix formation and therefore the resistance of the biofilm [3]. C2, C5, and C8 strains formed a biofilm whose composition in living bacteria was very similar, close to 90%. Only the C18 strain had a lower proportion with 70% live bacteria. This could be explained by the presence of large aggregates which could lead to bacteria death [39]. From a global point of view, the percentage of living bacteria within biofilm was higher in the dynamic than in the static model with the exception of C18 strains; with an average of 88.5% (±1%) for C2, C5, and C8 vs. an average of 56.8% (±9%) in the static model (data from reference [38]).

We compared biofilm structures of C2, C5, C8, and C18 after osteoblast infection in our dynamic model. They were similar to those formed before internalization for C5 and C8, but there were different for C2 and C18 (Figure 3a). Biofilm volumes were on average 1.5 times smaller after internalization than before (Figure 3b) except for C8. However, the C2 and C18 strains formed the most diminished biofilms with an almost three-fold decrease. Following the internalization, an increase in the proportion of dead bacteria was observed for all strains except for C18 in the dynamic model (Figure 3c). Biofilms formed in the dynamic model by the C5, C8, and C18 strains showed a rate of living bacteria around 80 ± 20%. We observed a heterogeneous distribution of C2 bacteria within biofilm with aggregates and also bacteria free-areas. The C2 strain showed a 50% loss of total biofilm volume after internalization (Figure 3b); however, there was a three-fold decrease of living bacteria volume and a 1.8-fold increase of dead bacteria volume. However, the rates were very variable, with a living bacteria proportion around 56 ± 40%. We noted that some aggregates were only composed of dead bacteria. We noticed an increase in biomass for C8 strain (Figure 3b) with a strong increase in dead bacteria proportion (×3) (Figure 3c). A similar behavior was observed for the C5 strain (20% loss of total biofilm with 2.3-fold increase of dead bacteria volume). Concerning the C18 strain, we evaluated a biomass decrease in equivalent proportions for live and dead bacteria (60–70% decrease). Unlike before internalization, we did not see any important mushroom-like structures for C18, neither a monolayer composed of an equal proportion of live and dead bacteria but a simple monolayer biofilm with some dead bacteria, similar to a C8 biofilm structure. All these results combined highlighted the important differences between *C. acnes* strains in terms of biofilm responses following internalization.

These data were compared to results previously obtained in a static model [38] to assess the modification of live/dead bacteria proportions after internalization in the two tested models. Even if the internalization of the four selected commensal strains (C2, C5, C8, and C18) had an impact in the biofilm formation through both models, the results clearly were in opposition. Indeed, our previous publication using the static model demonstrated an increase of live bacteria proportion of the four commensal *C. acnes* biofilm after internalization (×2.04 ± 0.53 in average). In the current dynamic model, we observed no important change except an increase in dead bacteria proportion for C2 strain (×4), and no volume expansion.

The explanation of the opposite results about live and dead biofilm proportion in the two models was the presence of variable conditions (oxygen and nutrient concentrations) as described above. Moreover, in static model, the dispersed bacteria remained close to biofilm whereas in dynamic models, dispersed bacteria were eventually carried away by the flow. The data presented here confirmed that all variations in biofilm condition culture will affect the results of a study [24]. In a bone context, such a flow and oxygen are limited, unlike infections on heart valves for example, underlining that a dynamic model to study biofilm in on prosthesis and joint infection context may not be as relevant as a static model.

To complete this approach, we also observed the biofilm formation of nine clinical strains isolated from bone and prosthesis infections (BPI). Biofilms formed in a dynamic model presented very heterogeneous structures (Figure 4). BPI strains 1, 5, 7, 8, and 9 developed very thick biofilms, the last three of which have very large aggregates, resembling the mushroom-type form and emphasizing a certain biofilm maturity (Figure 4a,b).

On average, BPI biofilms in flow cells models had a total volume of 2.8 × 10^6^ ± 0.6 UA while the commensal strains were on average twice as small (1.3 × 10^6^ ± 0.4 UA). However, some strains, such as BPI 2 or BPI4, showed a very fine biofilm, with only a few aggregates. In our previous publication [38], we have observed that BPI biofilms were more important than commensal ones before internalization. In this case, our results were concordant. Concerning the live/dead proportion, BPI biofilms in dynamic models were composed of 69 to 97% of live bacteria with an average of 89.5 ± 2.7% (Figure 4c), which is higher than in static models (61.2%) [38]. Again, the proportion of live bacteria was different in the two models confirming that permanent supply of a “fresh” medium with nutrients and oxygen in the flow cell model should allow better survival of bacteria within biofilm. To verify this hypothesis, we will have to repeat experiments in controlling these parameters.

## 4. Conclusions

In conclusion, in a bone and prosthesis infection context, liquid flow and oxygen are limited; therefore, a dynamic model to study *C. acnes* biofilm may not be as relevant as a static model. All these results underline the importance of the model choice for investigations on biofilm [23,24,40]. First, the medium in which biofilm is formed. In bone context, we described many different factors that could influence bacteria: lack of oxygen, nutrient starvation, excess of magnesium, and secretome of active bone cells [28]. Second, the presence of various types of surfaces will also affect bacteria attachment: bone surface, host matrix, and prosthesis implant, which is titanium most of the time [3]. For example, *C. acnes* biofilm is boosted on titanium surface [38].

There is an urgent need of in vitro biofilm model adapted to the clinical context to better understand mechanisms of biofilm infection to target specific biofilm characteristic but also to enhance antibiofilm molecules screening. Without consensus of in vitro biofilm models, in vivo biofilms remain the most representative methods to mimic infections in clinical context [41] especially in BPIs, where it is quite expensive.

## Figures and Tables

**Figure 1 microorganisms-09-02035-f001:**
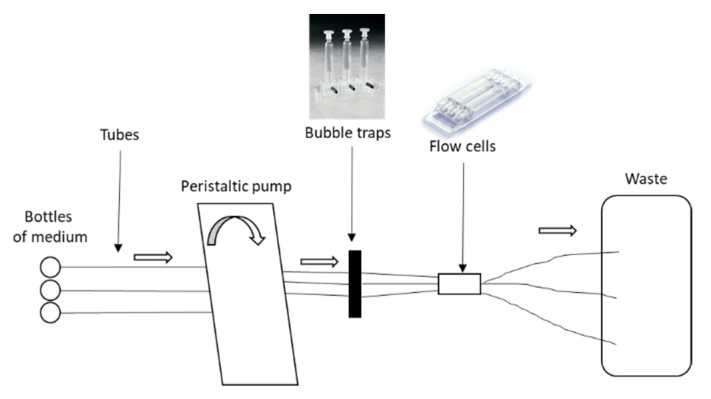
Flow cells model of *Cutibacterium acnes* biofilm.

**Figure 2 microorganisms-09-02035-f002:**
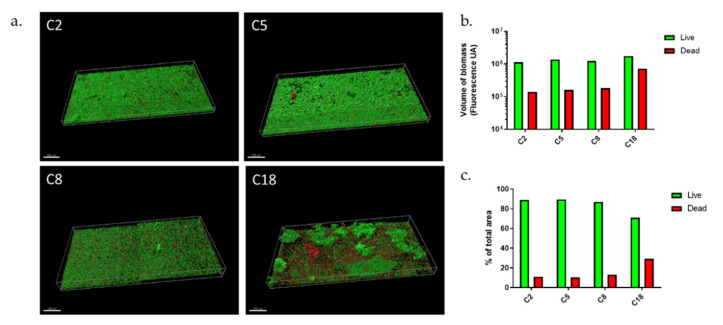
*C. acnes* biofilms in dynamic model. (**a**) Biofilm structure of commensal *C. acnes* strains marked with Syto9 (green) and PI (red) fluorochromes; (**b**) volume of biofilm biomass with repartition of Syto9 and PI staining; (**c**) repartition of Syto9 and PI staining in percentage (Acquisition of images by fluorescent microscopy of two joined fields of one sample and calculation by Imaris software). Scale = 100 µm.

**Figure 3 microorganisms-09-02035-f003:**
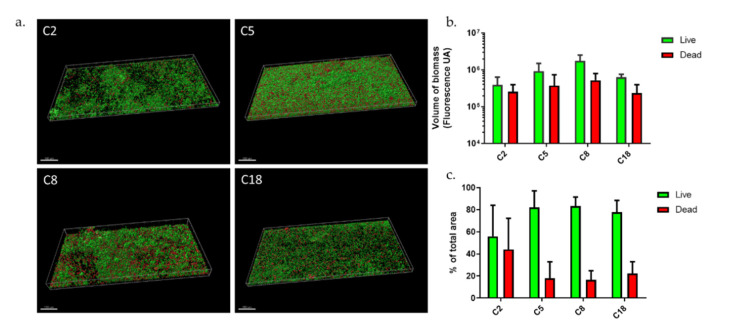
*C. acnes* biofilms after osteoblast-like internalization (**a**) Biofilm structure of commensal *C. acnes* strains in dynamic model; (**b**) volume of biofilm biomass with repartition of Syto9 and PI staining; (**c**) repartition of Syto9 and PI staining in percentage (Acquisition of images by fluorescent microscopy of two joined fields of one sample and calculation by Imaris software). Scale = 100 µm.

**Figure 4 microorganisms-09-02035-f004:**
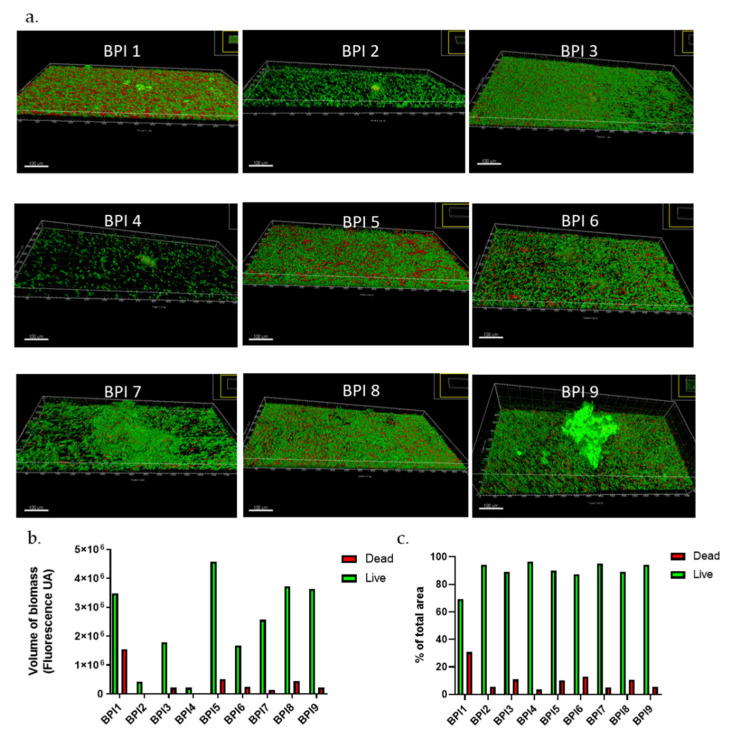
*C. acnes* biofilms of BPI strains in dynamic model. (**a**) Biofilm structure of commensal *C. acnes* strains marked with Syto9 (green) and PI (red) fluochromes; (**b**) volume of biofilm biomass with repartition of Syto9 and PI staining; (**c**) repartition of Syto9 and PI staining in percentage (Acquisition of images by fluorescent microscopy of two joined fields of one sample and calculation by Imaris software). Scale = 100 µm.

## Data Availability

Not applicable.

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
