# Peer review of "Comparison of Two Cutibacterium acnes Biofilm Models"

_microorganisms, 2021, doi:10.3390/microorganisms9102035_

Round 1

Reviewer 1 Report

Current study by Jennifer Varin-Simon and colleagues focuses on selection of a relevant biofilm model in accordance with the environmental or clinical context, to effectively improve the understanding of biofilms and the development of better antibiofilm strategies. Also in context of bone and prosthesis infection  a static model should be better for C. acnes biofilm. The study design is straightforward, and it is a well written article.

There are few minor concerns as follows:

In figure 3 legend, “(Acquisition of images by fluorescent microscopy and calculation by Imaris software).” statement should be modified (should be consistent with) like figure legends in figure 2 and 4.

“Figure 1: Flow cells model of Cutibacterium acnes biofilm.” Should be moved to result and discussion section in continuation with other figures.

Subfigures a, b in figure 1,2, 3 font size should be same.

Author Response

We thank the reviewer for the comments.

In figure 3 legend, “(Acquisition of images by fluorescent microscopy and calculation by Imaris software).” statement should be modified (should be consistent with) like figure legends in figure 2 and 4.

 We modified the legend of Figure 3 to be consistent with other legends of Figure 2 and 4.

“Figure 1: Flow cells model of Cutibacterium acnes biofilm.” Should be moved to result and discussion section in continuation with other figures.

 We moved the figure 1 in results and discussion section

Subfigures a, b in figure 1,2, 3 font size should be same.

We modified the font size of subfigures a, b in figures.

Reviewer 2 Report

The authors improved the manuscript according to the previous version and the paper is now in conditions to be accepted.

Author Response

We thank the reviewer for reviewing our paper.

This manuscript is a resubmission of an earlier submission. The following is a list of the peer review reports and author responses from that submission.

Round 1

Reviewer 1 Report

The communication entitled “Comparison of two Cutibacterium acnes biofilm models” by Varin-Simon et al., intends to evaluate and compare static and dynamic models of biofilms. the communication is very interesting and actual, being in line with the journal scopus. The English is good. The M&M section is well described and the results well explained and present. In my opinion the manuscript should be accepted after the follow minor concerns:

define in the first mention BPIs

line 215 – remove extra space

Reviewer 2 Report

In the current study Jennifer Varin-Simon and colleagues provided significant insight into selection of a relevant biofilm model in accordance with the environmental or clinical context, to effectively improve the understanding of biofilms and the development of better antibiofilm strategies. Authors have compared the impact of four selected commensal strains of C. acnes (C2, C5, C8 and C18) in the biofilm formation and modification of live/dead bacteria or C. acnes proportion after internalization through static and dynamic models. The study design is straightforward, and it is a nicely written article. I have few comments

More background on C. acnes and how different components of C. acnes plays a major role in adherence to skin tissue and for biofilm formation. Check previous publications…Whale GA et al, 2004 and Lee WL, 1982

In method section lacking of human IRB approval for C. acnes strains, commensal and clinical isolates from BPIs

There are many bacteria like Enterobacteriaceae, Pseudomonas, staphylococci, Coryneform, pneumococci, fungi, yeasts and Candida species also grow on Columbia agar or Brain Heart Infusion (BHI) broth with 5% sheep blood. In this case, how C. acnes selection in these media was done? Did author also confirmed these C. acnes strains through biochemical tests such as catalase and indole test?

A representative image of Flow cells models should be included in this study for better understanding.

In line 119 of result and discussion section author stated that,” As for the C18 strain, the formed biofilm was completely different with large, very thick aggregates, mainly composed of live bacteria”. However, the proportion of live bacteria in C18 is lesser than C2, C5 and C8 strains. The statement should be modified.

Did author tested sensitivity of in vitro-grown C. acnes  biofilms to a number of relevant antibiotics.

In figure 1b and c, C. acnes biofilms in dynamic model, error bars should be included in graphs from independent experiments.

Further genomic study should be done to find link between the commensal and clinical C. acnes strains from BPI and phylotype which may provide significant finding on involvement of specific phylotype in Biofilm formation in BPI

In line 180 of result and discussion section authors states that, “In the current dynamic model, we noticed a decrease in dead bacteria proportion for C2 and C18 180 strains.”. However, an increase in dead bacteria proportion for C2 was detected in Figure 2C.

In Figure 3. C. acnes biofilms of BPI strains in dynamic model, errors bar should be provided for graphs in 3b and 3c from independent experiments.